# Intestinal Biomarkers in Preterm Infants: Influence of Mother’s Own Milk on Fecal Calprotectin and of Gestational Age on IFABP Concentrations

**DOI:** 10.3390/nu17132177

**Published:** 2025-06-30

**Authors:** Carla Balcells-Esponera, Victoria Aldecoa-Bilbao, Cristina Borràs-Novell, Miriam López-Abad, Anna Valls Lafón, Marta Batllori Tragant, Montserrat Izquierdo Renau, Beatriz del Rey Hurtado de Mendoza, Ana Herranz-Barbero, Isabel Iglesias-Platas

**Affiliations:** 1Neonatology Department, BCNatal—Centre de Medicina Maternofetal i Neonatologia de Barcelona, Hospital Sant Joan de Déu, Universitat de Barcelona, 08950 Barcelona, Spain; miriam.lopez.abad@gmail.com (M.L.-A.); mizquierdo@hsjdbcn.org (M.I.R.); beatriz.reyhurtado@sjd.es (B.d.R.H.d.M.); isabel.iglesias@sjd.es (I.I.-P.); 2Neonatology Department, BCNatal—Centre de Medicina Maternofetal i Neonatologia de Barcelona, Hospital Clínic, Universitat de Barcelona, 08028 Barcelona, Spain; valdecoa@clinic.cat (V.A.-B.); cborrasn@clinic.cat (C.B.-N.); herranz@clinic.cat (A.H.-B.); 3Biochemistry Department, Hospital Sant Joan de Déu, 08950 Barcelona, Spain; annafrancia.valls@sjd.es (A.V.L.); marta.batllori@sjd.es (M.B.T.); 4Neonatal Intensive Care Unit, Norfolk and Norwich University Hospital, Norwich NR4 7UY, UK

**Keywords:** fecal calprotectin, intestinal fatty acid-binding protein, intestinal maturation, preterm birth

## Abstract

**Background/Objectives:** Calprotectin and intestinal fatty acid-binding protein (IFABP) may reflect the intestinal maturation process of very preterm infants (VPI) but have also been associated with gut inflammation. To establish normative values for fecal calprotectin (FC) and urinary intestinal fatty acid-binding protein (uIFABP) in VPI and to study their correlations with demographic and clinical factors. **Methods:** A cohort of VPI (born before or at 32.0 weeks of gestation) was recruited in two neonatal intensive care units. Urine and fecal samples were collected at 1, 4 and 8 weeks of life to measure urinary IFABP (normalized to creatinine as uIFABP/Cr) and FC, respectively. UIFABP was determined by ELISA and FC by fluoroenzyme immunoassay. **Results**: 194 newborns had at least one valid biomarker measurement. The study cohort mean gestational age was 28.9 ± 2.3 weeks and mean birth weight 1178 ± 365 g. Although uIFABP/Cr concentrations differed between the two centres, they were negatively correlated with gestational age, with a statistically significant correlation observed in both centres at week 4 (Hospital Clínic: Spearman’s rho −0.500; *p* = 0.000 and Hospital Sant Joan de Déu: Spearman’s rho −0.474; *p* = 0.000). Conversely, FC showed a positive significant correlation at the same time point (Spearman’s rho 0.302; *p* = 0.006). At week one, FC increased with antibiotic exposure (28 mcg/g of stool per antibiotic day, 95%CI 3–57; *p* = 0.028). FC at week 4 was inversely correlated with mother’s own milk (MOM) exposure during the first month (Spearman’s rho −0.253; *p* = 0.023). **Conclusions**: uIFABP/Cr and FC are associated with gestational age at 4 weeks and FC is also influenced by antibiotic treatment and MOM exposure.

## 1. Introduction

The gastrointestinal tract plays an essential role in both nutrition and immune function during neonatal development. After very preterm birth, the maturation of the gastrointestinal tract occurs partially within the intensive care unit environment, which differs significantly from intrauterine conditions [1]. Although the anatomical differentiation of the human gut is completed by the 20th week of gestation, it is during the third trimester of pregnancy that the fetal intestine becomes structurally and physiologically prepared for extrauterine life [2].

Gut immaturity leads to a number of issues. A primary clinical sign in preterm infants is intestinal dysmotility, which can lead to feeding intolerance and delayed meconium passage [3]. Regarding immunological function, the fetal enterocyte possesses a high endocytic capacity, enabling the transfer of immune macromolecules from the amniotic fluid into the fetal circulation. Following preterm birth, this gut barrier remains permeable, allowing the passage of potentially harmful molecules from the intestinal lumen, which may trigger an exaggerated inflammatory response [4].

Mother’s own milk (MOM) is widely recognized as the optimal feeding choice for neonates. Beyond its nutritional properties, mother’s milk contains bioactive components that help attenuate the enterocyte inflammatory response [5]. Our research group has previously identified several bioactive peptides in MOM [6] that, among others [7], may contribute to its immunomodulatory and anti-inflammatory properties, potentially reducing the risk of necrotizing enterocolitis in very preterm infants [8].

The availability of non-invasive biomarkers for assessing gut maturity and inflammation would assist clinicians in identifying patients at greater risk of gastrointestinal disorders. Calprotectin is a calcium-binding protein found in the cytoplasm of neutrophils and serves as an indicator of neutrophil migration to the gastrointestinal tract when measured in stool [9]. In 2021, the Position Paper of the European Society for Paediatric Gastroenterology and Nutrition emphasized the role of fecal calprotectin (FC) as a useful inflammatory marker for the diagnosis and monitoring of inflammatory bowel disease in pediatric patients [10]. Additionally, a meta-analysis focusing on very preterm infants suggests that FC might also offer high diagnostic value in cases where necrotizing enterocolitis is clinically suspected [11].

Intestinal fatty acid-binding protein (IFABP) is a water-soluble peptide located in the cytoplasm of the enterocyte. It is released into the bloodstream in response to intestinal cell damage and is subsequently excreted in urine after a brief plasma half-life [12]. Plasmatic concentrations of this biomarker are typically very low in healthy individuals [13] and may be correlated with the severity of gastrointestinal damage in preterm newborns with necrotizing enterocolitis [14].

Despite a growing interest in the potential utility of intestinal markers for evaluating neonatal gut integrity, limited data exist on the levels of FC and uIFABP in very preterm infants. The aim of this study was to contribute to the generation of normative data for FC and uIFABP levels in a cohort of very preterm infants. Additionally, we examined neonatal and postnatal factors that may influence the variability in their concentrations.

## 2. Materials and Methods

### 2.1. Study Design

This observational cohort study was conducted at Hospital Sant Joan de Déu and Hospital Clínic, two tertiary care centres providing neonatal care within BCNatal (Barcelona Center for Maternal-Fetal and Neonatal Medicine), adhering to shared guidelines to ensure homogeneous clinical care.

### 2.2. Study Population

From January 2018 to April 2020, very preterm infants (born at or before 32.0 weeks of gestation) admitted to the neonatal units of the participating centres before the 7th day of life were eligible for participation if their mothers intended to breastfeed. Exclusion criteria included congenital malformations, known chromosomal, genetic or metabolic abnormalities, and low short-term survival prospects as assessed by the attending clinical team. Both the local research ethics committee approved the protocol (PIC-147-17 and HCB/2016/0959) and written informed consent was obtained from parents or legal guardians prior to inclusion.

### 2.3. Sampling and Laboratory Procedures

**Urinary IFABP:** Urine was collected by placing a cotton ball in the diaper adjacent to the urethral meatus to minimize fecal contamination. A 1–2mL sample was then frozen at −80 °C for later batch analyses. IFABP concentration was determined with a commercial ELISA kit (Human FABP2/I-FABP Immunoassay, Quantikine^®^ ELISA, R&D, Minneapolis, USA), and normalized against creatinine (uIFABP/Cr) to account for variations in urine concentration [15]. Samples were analyzed in duplicate, and results with a coefficient of variation > 15% were excluded.

**Fecal Calprotectin**: Stool samples, approximately the size of a rice grain, were collected from the diaper using a sterile pipette and frozen at −80 °C for further analyses. FC was measured using a fluoroenzyme immunoassay (EliA Calprotectin 2 well. Thermo Fisher Scientific Inc.^®^, Waltham, MA, USA). The extraction of calprotectin from feces was performed using an extraction buffer (EliA Stool Extraction Buffer plus. Thermo Fisher Scientific Inc.^®^, Massachusetts, USA) weighing on a precision balance between 100 and 200 mg of stool and adding the extraction buffer following the formula: **x** mg stool x 85 = **y** mcL EliA Stool Extraction Buffer plus. The results were expressed in mcg/g feces.

Both urinary and fecal samples were collected at 1, 4 and 8 weeks of life.

**Mother’s milk hormones**: In the same cohort of very preterm infants, mother’s milk concentrations of leptin, insulin, adiponectin and milk fat globule endothelial growth factor-8 (MFG-E8) at week 4 have been previously analyzed and the results reported elsewhere [6].

All samples were analyzed in batches to minimize variability.

### 2.4. Clinical Data

Neonatal clinical data were obtained from medical records, and infants were followed until discharge. Intrauterine growth restriction was defined as estimated fetal weight below the 3rd centile or below the 10th centile with abnormal Doppler findings, while small for gestational age was defined as a birth weight below 10th centile for gestational age [16].

Days on intensive care were defined according to the criteria of level 1 of the British Association of Perinatal Medicine [17].

**Feeding protocol**: Parenteral nutrition was initiated as soon as possible following admission and establishment of venous access. Enteral nutrition commenced, in stable patients, as soon as the maternal colostrum was available, or, if it was not, after the parents granted consent for the use of donor milk. According to the local protocol, enteral nutrition was initiated at 20–35 mL/kg/day as soon as possible after birth if the patient was clinically stable, with a progression of 20–30 mL/kg/day depending on gestational age—the lower range being used for the more immature and IUGR infants. Total enteral nutrition was defined as enteral intakes exceeding 120 mL/kg/day [18]. Fortification with a bovine-derived multicomponent fortifier (Pre-NAN FM85^®^, Nestle^®^, Vevey, Switzerland) was started upon reaching an enteral intake of 80 mL/kg/day of human milk according to current guidelines on enteral nutrition [19]. Data were collected on both total volume intake and the amount of each type of milk received. The proportion of MOM received during the first month was calculated as the volume of MOM in relation to total enteral intake.

Glycerol enemas were administered every 12–24 h starting at 48 h of life in the absence of spontaneous meconium. Delayed meconium evacuation was defined as the absence of meconium passage within 72 h of life [20].

**Growth data**: Weight was recorded daily, and length and head circumference were measured weekly as part of routine clinical care. Growth during admission was assessed in relation to the birth weight curves generated by the Intergrowth-21st project [21]. The initial weight loss was calculated as the ratio between the lowest recorded weight during admission and birth weight.

Postnatal growth failure was defined as a fall in weight, length or head circumference z-score (ZS) between two time points greater than 2 [22,23].

**Neonatal outcomes**: Every patient was classified to have an early or late onset sepsis (EOS/LOS) if a bacterial pathogen was isolated in blood and/or cerebrospinal fluid culture before/on or after day 3, respectively [24]. Patients with treated PDA (patent ductus arteriosus) were those that received ibuprofen and/or acetaminophen to actively close a clinically significant PDA. Necrotizing enterocolitis (NEC) was defined according to Bell’s criteria [25] or based on the evidence of gut perforation during surgery. A mature stooling pattern was defined as at least 2 spontaneous bowel movements within 24 h over three consecutive days [26]. Bronchopulmonary dysplasia was defined as the need for oxygen supplementation for at least 28 days [27]. Retinopathy of prematurity (ROP) was classified according to the International Committee for the Classification of ROP [28], and intraventricular hemorrhage was graded according to Papille’s classification [29].

All data were collected and managed using a database specifically created for this study, employing REDCap^®^ v14.3.0 electronic data capture tools hosted at Hospital Sant Joan de Déu, within the secure REDCap platform [30].

### 2.5. Statistical Analysis

A descriptive analysis was conducted to estimate FC and urinary IFABP concentrations in very preterm infants at 1, 4 and 8 weeks of life. Additionally, a case–control analysis was performed to identify factors associated with biomarker levels, and a prospective longitudinal analysis examined the relationships between intestinal biomarkers, in-hospital growth, and clinical outcomes of these very preterm infants.

Statistical methods were employed to estimate concentrations of intestinal biomarkers in the preterm population. Relationships between quantitative variables were analyzed using correlation methods (Pearson or Spearman, depending on variable distribution). Differences between groups were assessed using Student’s *t* or Mann–Whitney U tests, and comparisons of biomarkers across weeks were conducted using paired samples t-test. The distribution of qualitative variables between groups was analyzed using chi-square tests, with Fisher’s exact correction when appropriate for sample size. Results with a *p*-value < 0.05 were considered statistically significant. Evaluation of confounders was performed using multivariate regression analysis.

## 3. Results

### 3.1. Sample Description

During the study period, 227 newborns born at or before 32 weeks of gestation and their 195 mothers were recruited. These analyses are based on a sample of 194 newborns that had at least one valid intestinal marker measurement. Demographic data, nutritional details and key clinical outcomes are summarized in Table 1.

No correlation was found between uIFABP/Cr and FC concentrations. FC values at the studied time points were not correlated; however, there was a significant positive correlation between uIFABP/Cr at week 1 and week 4 (Spearman’s rho 0.432; *p* = 0.001) and between uIFABP/Cr at week 4 and week 8 (Spearman’s rho 0.754; *p* = 0.001).

uIFABP/Cr levels were consistently and significantly higher in patients admitted to Hospital Clínic compared to those born at Hospital Sant Joan de Déu at all three time points (Appendix A. Table A1); consequently, all subsequent analyses of uIFABP/Cr were adjusted for the centre, and correlations involving data from week 8 are not shown due to a small remaining sample size.

Postnatal dynamics of FC, stratified by gestational age, are shown in Table 2.

### 3.2. Associations Between Intestinal Markers, Gestational Age and Neonatal Variables

Neither uIFABP/Cr nor FC was associated with prenatal conditions. UIFABP/Cr levels showed a negative correlation with gestational age, significant in both centres at week 4 (Figure 1).

FC had a weak positive correlation with gestational age at 4 weeks (Figure 2).

No significant correlation was observed between uIFABP/Cr and neonatal morbidities after adjusting for centre and gestational age (Table A2). FC levels at week 1 increased with antibiotic exposure; specifically, for each day of antibiotic use, FC increased by 28 mcg per g of stool (95%CI 3–57); *p* = 0.028. There was a correlation between days of antibiotic exposure and fecal calprotectin levels at week 1, although we could not find a cut-off point for a number of days on antibiotics that determined a significant difference in calprotectin levels between groups (Figure 3). In adjusted analyses, neither uIFABP/Cr nor FC showed an association with neonatal growth during admission.

### 3.3. Impact of Enteral Nutrition on Intestinal Biomarker Concentrations

FC at week 4 exhibited a negative correlation with exposure to MOM during the first 4 weeks of life, with a significant decrease in infants for whom MOM constituted more than 62% of enteral feeds and reaching a minimum in preterm neonates with an exclusive MOM diet (Figure 4). Exposure to MOM did not have an impact on uIFABP/Cr levels. The total amount of enteral nutrition received during the first month did not affect the levels of the intestinal biomarkers.

Both intestinal biomarkers were partially related to maternal milk adiponectin concentration. The relationship between milk adiponectin and uIFABP/Cr was positive, with Pearson’s r values of 0.319 (*p* = 0.062), 0.118 (*p* = 0.273) and 0.569 (*p* = 0.043) at weeks 1, 4, and 8, respectively. In contrast, FC showed an inverse correlation with milk adiponectin that was statistically significant at week 1 (Pearson’s r –0.373; *p* = 0.012). The correlations for weeks 4 and 8 were as follows: week 4, Pearson’s r –0.259 (*p* = 0.09) and week 8, Pearson’s r 0.005 (*p* = 0.983). No significant association was found between insulin, leptin or MFG-E8 levels and the intestinal biomarkers studied.

## 4. Discussion

This study aimed to provide normative values for FC in very preterm infants at 1, 4 and 8 weeks of life. Our findings demonstrate a decrease in uIFABP/Cr levels with increasing gestational age and a lower concentration of FC at week 4 in neonates who received a higher proportion of MOM.

Data on normative values of intestinal biomarkers in preterm infants remain limited. Shores et al. [31] reported serum IFABP and serum calprotectin concentrations in a cohort of neonates born between 24 and 40 weeks of gestation. Our sampling of urine and feces, however, has the advantages of being less invasive and more feasible in clinical practice [32], allowing for sequential measurements without contributing to anemia.

As previously reported, fecal calprotectin levels after preterm birth are elevated during the first days of life, decrease during the second week, and then rise again thereafter [33].

In line with previous reports, uIFABP/Cr showed a negative correlation with gestational age [34,35], which may reflect increased enterocyte turnover during early gut maturation. In contrast, Shores et al. [12,31] did not find differences in IFABP concentrations across gestational ages when measurements were performed in serum. Given the quick renal clearance and very short plasmatic half-life [36] of fatty acid-binding proteins (a group of small proteins), it has been suggested by other authors [37] that urinary IFABP may have greater accuracy than serum IFABP in the prediction of intestinal damage.

Thai et al. [34] did not identify significant differences in FC levels across patient groups with varying gestational ages. This suggests that it may be not feasible to establish a definite FC cut-off point based on maturity at birth. In our analysis, when gestational age was treated as a continuous variable, we observed a significant positive correlation between FC at 4 weeks and gestational age. However, when categorizing by gestational age groups, FC appeared to be similar between groups. Consistent with our findings, Shores et al. [31] observed higher FC levels at greater gestational ages during the first week of life in a cohort of preterm infants.

MOM is known to contain several bioactive factors that promote an anti-inflammatory state in the neonatal bowel [38]. Some authors have reported higher FC levels in exclusively breastfed neonates [39,40], suggesting enhanced bowel maturation compared to mixed- or formula-fed infants. Conversely, and consistent with our findings, recent data indicate a negative correlation between MOM exposure and FC levels in preterm infants [41,42,43,44], suggesting a protective effect of MOM against gastrointestinal inflammation when compared to both donor milk and formula. Our findings in a cohort of very preterm infants with a high prevalence of exclusive MOM feeding indicate a potential dose-dependent effect, where higher percentages of MOM intake would be associated with lower levels of inflammatory markers.

In analyzing mother’s milk, adiponectin was found to be positively correlated with uIFABP/Cr and inversely associated with FC. To our knowledge, no prior publications have focused on the impact of maternal milk hormone content on intestinal maturation and inflammation. Although these results warrant further research before definitive conclusions can be drawn, our data suggest that bioactive metabolites in mother’s milk may play specific roles in the maturation of the preterm gut.

Regarding NEC, previous studies had highlighted the high sensitivity and specificity of FC in this condition [11,45], while also noting its wide inter- and intra-individual variability in neonates [46]. This variability complicates the establishment of cut-off points for predicting and diagnosing gastrointestinal disorders. Another potential area of research for the application of intestinal biomarkers would the guidance for re-introduction of enteral nutrition after NEC [47,48]. Some data, including a recent meta-analysis [49], suggest that uIFABP/Cr may be an accurate marker for the prediction, diagnosis and severity grading of NEC [45,50,51,52]. However, a systematic review including 14 published reports [53] concluded that sensitivity is low, which may be partially explained by wide methodological heterogeneity. Our study was not designed to address this question, and in our cohort of preterm neonates with pre-specified sampling times, patients with NEC, isolated intestinal perforation or immature stooling patterns had comparable biomarker concentrations to their healthy counterparts. Based on our results, further research on the clinical applicability of FC and uIFABP/Cr in neonatal patients with gastrointestinal disorders is needed, but future study designs must account for major confounders, particularly gestational age and MOM exposure.

Our analysis also highlights that antibiotic exposure is associated with higher FC levels at week 1. While some studies suggest a protective role of antibiotics against NEC development [54], the patients in those studies had significantly higher early antibiotic exposure than those in our cohort. In line with our findings, previous data have indicated a higher risk of NEC in patients receiving prolonged antibiotic treatments [55]. Although we could not establish a link between FC concentrations and NEC, the increase in early FC levels with longer antibiotic therapy may suggest a heightened proinflammatory state in patients undergoing antimicrobial treatment, possibly due to the impact of antibiotics on increasing intestinal permeability [56].

Even though we included some potential confounding factors in our analyses, there may be others—such as maternal health status or the patient’s condition at birth—that could also have influenced biomarker concentrations. We chose to include a targeted set of variables in order to limit the risk of increasing type I error and false positive associations.

A significant limitation of this study is the unexpected difference in uIFABP/Cr concentrations (with comparable urinary creatinine levels) between the two centres. This discrepancy may result from methodological issues or genuine clinical differences. The sample acquisition procedures were reviewed several times and found to be identical. Furthermore, to minimize variability between experiments, samples were analyzed in batches. There may, however, still be methodological factors that have gone undetected, as Staub et al. have recently pointed out [57]. If the reported concentration differences are clinically significant, they may reflect a confounder that we have not been able to control. This inter-centre variability in IFABP levels—despite the absence of an identifiable methodological or clinical explanation—may provide relevant information for other research groups investigating IFABP in preterm infants. In light of these findings, we did not report normalized UIFABP/Cr data. Nevertheless, the stratified analysis reveals a similar pattern in UIFABP/Cr concentrations in relation to the studied variables across the individual centres, which supports the validity of the results obtained.

In the absence of clear criteria to define healthy gastrointestinal function in preterm infants and given that no differences were observed between patients with and without NEC, we included the uIFABP/Cr and FC values of the entire cohort in our analysis.

## 5. Conclusions

UIFABP/Cr and FC can be non-invasively measured in preterm infants, and this study provides valuable insights into their normative values. UIFABP/Cr in very preterm infants is inversely correlated with gestational age, while MOM exposure is negatively correlated with FC in a dose-dependent manner. Both uIFABP/Cr and FC might prove useful in the diagnosis and management of gastrointestinal conditions in the neonatal period, but further research is required to better understand their physiological significance and the factors influencing their levels before considering clinical implementation.

## Figures and Tables

**Figure 1 nutrients-17-02177-f001:**
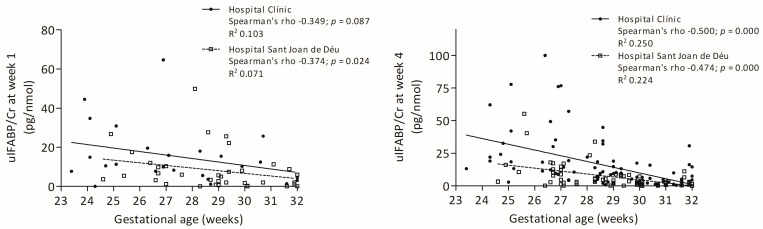
Correlations between gestational age and uIFABP/Cr levels at weeks 1 and 4.

**Figure 2 nutrients-17-02177-f002:**
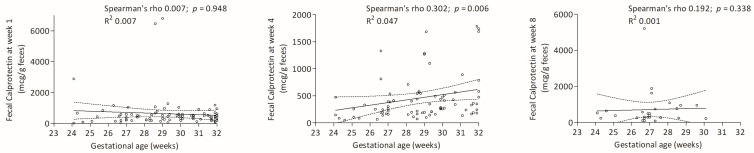
Correlations between gestational age and FC levels at weeks 1, 4 and 8.

**Figure 3 nutrients-17-02177-f003:**
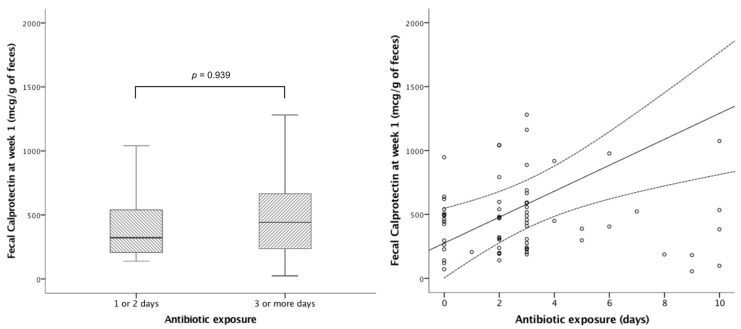
Impact of early antibiotics on fecal calprotectin at week 1. Left panel: Stratification of antibiotic treatment into 1 or 2 days (median fecal calprotectin 321 mcg/g of feces (IQR 203−569)) vs. 3 or more days (median fecal calprotectin 434mcg/g of feces (IQR 230−611)) *p* = 0.939. Right panel: Correlation between antibiotic exposure and fecal calprotectin at week 1 (R^2^ 0.128; *p* = 0.002).

**Figure 4 nutrients-17-02177-f004:**
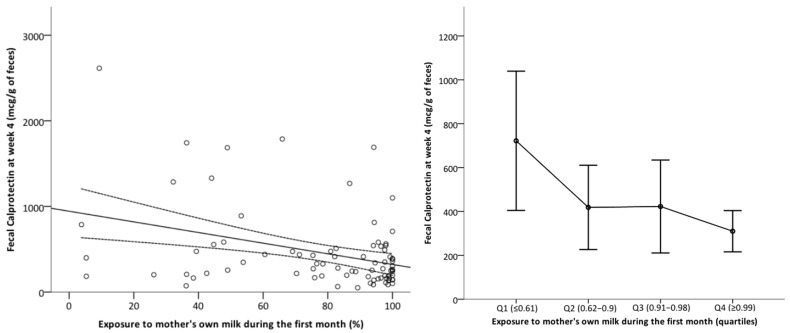
Impact of exposure to mother’s own milk during the first month of life on fecal calprotectin levels at 4 weeks. Left panel: Scatterplot with regression line. Y-axis: fecal calprotectin levels at week 4 (mcg/g of stool). X-axis: % of mother’s own milk (MOM) in the total enteral feeding received up to 28 days by the infant. Right panel: Mean fecal calprotectin at week 4 by quartiles of MOM exposure. Error bars represent the confidence interval 95%.

**Table 1 nutrients-17-02177-t001:** Demographics, nutrition and main outcomes of the study cohort.

**Prenatal Condition**		**Neonatal Characteristics**	
IUGR	37 (19.1)	Female sex	100 (51.5)
Histologic Chorioamnionitis	52 (26.9)	Gestational age (w)	28.9 ± 2.3
Prenatal steroids (≥2 doses)	161 (83.0)	Birth weight (grams)	1178 ± 365
C-section	95 (49.0)	SGA	27 (13.9)
**Postnatal Morbidities**		**Neonatal Nutrition and Growth**	
Surfactant	57 (29.8)	Total enteral feeds (DOL)	8.8 ± 4.6
Postnatal steroids	17 (8.8)	% Initial weight loss	−9.7 ± 4.9
BPD	63 (33)	PN (days)	10.0 ± 9.3
IVH ≥ 2	5 (2.6)	% MOM at 28 days	78 ± 26
ROP > 2	12 (6.2)	% MOM at 36w PMA	77 ± 35
Treated PDA	33 (17.0)	Change W-ZS birth to 36w PMA	−0.99 ± 0.86
EOS	5 (2.6)	Change L-ZS birth to 36w PMA	−1.18 ± 1.3
LOS	39 (20.0)	Change HC-ZS birth to 36w PMA	−0.15 ± 1.1
NEC or SIP	17 (8.8)	PGF (weight) at 36w PMA	29 (13.7)
Death before discharge	4 (2.1)	PGF (length) at 36w PMA	44 (43.8)
PMA discharge (w)	37 ± 2.5	PGF (head circumference) at 36w PMA	12 (11.5)

Values are expressed as number (percentage) or mean ± standard deviation.

**Table 2 nutrients-17-02177-t002:** Postnatal dynamics of fecal calprotectin according to gestational age. Results are presented divided by gestational age group (<28/≥28 weeks at birth) and by chronological age at sample collection (under “week”). N = number of samples analyzed at each time point. Patients could contribute samples at 1 or more time points.

Gestational Age	Week	n	Fecal Calprotectin (mcg/g feces)				
			Median	IQR	Q1	Q2	Q3
23^0^–27^6^	w1	23	422	464	201	422	665
	w4	27	250	300	157	250	457
	w8	21	383	501	248	383	749
28^0^–32^0^	w1	64	449	310	286	449	595
	w4	54	349	363	215	349	578
	w8	6	578	---	210	578	---

Abbreviations: IQR: interquartile range; w: week.

## Data Availability

The datasets generated and analyzed during the current study are available from the corresponding author on reasonable request.

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
