# Peer review of "Intestinal Biomarkers in Preterm Infants: Influence of Mother’s Own Milk on Fecal Calprotectin and of Gestational Age on IFABP Concentrations"

_nutrients, 2025, doi:10.3390/nu17132177_

Round 1

Reviewer 1 Report

Comments and Suggestions for Authors

In this manuscript, the authors investigated the relationships between intestinal biomarkers in VPI, specifically FC and uIFABP, and gestational age as well as MOM. The objective was to establish normative values for FC and uIFABP in VPI and to explore the correlations between these biomarkers and demographic and clinical factors. The results indicated that uIFABP/Cr was negatively correlated with gestational age, whereas FC at week 4 was negatively correlated with the amount of MOM exposure. This study is relatively complete and has led to interesting conclusions. However, there are some suggestions that need to be addressed before accepting it for publication.

  1. There are no line numbers in the manuscript, which is not friendly to the reviewers.
  2. Where is the first reference? The reference numbers appearing in the main text of the original manuscript do not match those in the list of references at the end.
  3. The manuscript mentions the Spearman’s rho and statistical significance (p=0.001) in various places but does not provide effect sizes, which would be useful to evaluate the practical significance of the observed differences.
  4. The presentation method of data in the manuscript is rather limited, relying solely on scatter plots. While these can demonstrate certain results, they lack visual appeal and fail to engage readers effectively. An outstanding figure can significantly enhance the quality of a manuscript.
  5. When analyzing the associations between biomarkers and various factors, the authors have already considered some potential confounding factors, but there may be other unconsidered confounding factors (such as maternal health status, the infant's Apgar score at birth, etc.). The authors could mention these potential confounding factors in the discussion section and discuss their possible impacts on the results.
  6. The manuscript discusses the potential of FC and uIFABP/Cr as biomarkers for intestinal maturation and inflammation, but it does not fully explore the specific clinical application value of these biomarkers.
  7. The manuscript mentions that there are differences in uIFABP/Cr concentrations between the two research centers despite comparable urinary creatinine levels. The authors suggest that this could be due to methodological issues or genuine clinical differences, but they do not provide further explanation or analysis. It is recommended that the authors further explore the possible reasons for this discrepancy and conduct a more in-depth discussion in the discussion section.
  8. I think it is appropriate for the authors to add references in the methodology. Please thoroughly review and update the reference list to ensure the completeness and accuracy of each citation. In addition, it is recommended to replace outdated literature with literature that is less than three years old.

Author Response

Please see the attachment. There are figures and tables in the responses, this is why we did not use the suggested template. 

Reviewer 2 Report

Comments and Suggestions for Authors

Thank you for the opportunity to review this interesting article.
Here you are my comments.

Title: it should be not written in Capital letters. Report in the title also which biomarker was related (fecal calprotectin).

Methods: could you report about the enteral rate advancement? Was the same in the two centers? and the same within the infants (Beyond IUGR and GA)?
When you report postnatal growth definition please report loss of z-scores both for weight, length and head circumference and cite also Extra-uterine growth restriction in preterm infants: Neurodevelopmental outcomes according to different definitions. Eur J Paediatr Neurol. 2021 Jul;33:135-145. doi: 10.1016/j.ejpn.2021.06.004. 

In Table 1 please report also how many infants had EUGR/postnatal growth restriction according to weight, length and head cirmcuference loss of Z-scores.

I think you have to report data also for uIFABP/Cr for Table 2, Beyond the small sample size.

For table 2 i haven't understood why you reported both mean and percentiles. Had data a normal distribution or not?
If they had no normal distribution you have to report median and interquartile range only.
Why there is no p90 for w8 of 28-32 weeks?
Please report also data for all infants.

Please compare infants who 

Have you data to report about donor human milk? Did you use it?

Results: you report 194 newborns but the sum 23+27+21+64+54+6= 195 newborns. Please correct

Could you report also an image describing that for each day of antibiotic use, FC increased by 28
mcg per g of stool (95%CI 3-57); p=0.028

Reading the results, i would modify the title considering that both intestinal markers were not significantly associated with gestational age 
I would modify the title in "Fecal calprotectin is associated with mother's own milk exposure in preterm infants"
or "Fecal calprotectin is associated with mother's own milk exposure in preterm infants, whereas intestinal fatty acid-binding protein is not"

Report data on uIFABP as secondary outcome, and express the results primarily on fecal calprotectin.

Please comment why calprotectin was higher in w1, then decreased and returned higher again at w8.
Have you an image of dynamics, showing the dynamics were the same in the two subgroups?

Please report data on fecal calprotectin in infants before NEC or SIP (8.8% is not a low incidence)
if possible also data of FC when nutrition was reintroduced after NEC episode, and which type of milk was given

Discussion
I would change: This study "aimed" to provide.
Always change in reporting fecal calprotectin as primary outcome and the n uIFABP as secondary outcome

I would discuss the lack of data on FC and fortification of human milk (is not ethic give unfortified HM to preterm infants, considering that fortification is crucial to improve growth. you can cite Improving growth in preterm infants through nutrition: a practical overview. Front Nutr. 2024 Sep 10;11:1449022. doi: 10.3389/fnut.2024.1449022.)

I would discuss the lack of data on FC before and after NEC/SIP episode, considering that to date there is no evidence for optimal nutrition after NEC when human milk is unavailable
(you can cite 
Nutritional Strategies for Preterm Neonates and Preterm Neonates Undergoing Surgery: New Insights for Practice and Wrong Beliefs to Uproot. Nutrients. 2024 May 31;16(11):1719. doi: 10.3390/nu16111719.
and
Nutritional management after necrotizing enterocolitis and focal intestinal perforation in preterm infants. Pediatr Res. 2024 Jul 11. doi: 10.1038/s41390-024-03386-y.)

Author Response

Please see the attachment. We did not use the suggested template because tables and figures were difficult to fit in. 

Round 2

Reviewer 1 Report

Comments and Suggestions for Authors

The author addressed the questions I raised and improved the quality of the manuscript.